# Precision Medicine Approach Based on Molecular Alterations for Patients with Relapsed or Refractory Multiple Myeloma: Results from the MM-EP1 Study

**DOI:** 10.3390/cancers15051508

**Published:** 2023-02-28

**Authors:** Fabio Andreozzi, Matteo Dragani, Cyril Quivoron, Fabien Le Bras, Tarek Assi, Alina Danu, Karim Belhadj, Julien Lazarovici, Sophie Cotteret, Olivier A. Bernard, Vincent Ribrag, Jean-Marie Michot

**Affiliations:** 1Département d’Innovation Thérapeutique et d’Essais Précoces, Gustave Roussy, 94805 Villejuif, France; 2Hematology Department, Institute Jules Bordet, 1070 Bruxelles, Belgium; 3Hematology Department, Gustave Roussy, 94805 Villejuif, France; 4Translational Research Hematological Laboratory, Gustave Roussy, 94805 Villejuif, France; 5INSERM U1170, Université Paris-Saclay, Gustave Roussy, 94805 Villejuif, France; 6Hematology Department, Assistance Publique Hôpitaux de Paris, Hôpital Henri Mondor, 94000 Créteil, France; 7Department of Medical Biology and Pathology, Gustave Roussy, 94805 Villejuif, France

**Keywords:** personalized molecular therapies, multiple myeloma, next-generation sequencing

## Abstract

**Simple Summary:**

Personalized treatment for patients with relapsed or refractory multiple myeloma (r/r MM) remains an ongoing challenge, and there are no anti-myeloma therapies based on molecular abnormalities available. Identifying recurrent molecular abnormalities could allow for guiding patients toward appropriate targeted therapy. The MM-EP1 (Multiple Myeloma Early Phase −1) study aimed to assess whether patients who received molecularly oriented therapy may show improved outcomes. In our study, a molecularly oriented approach showed similar outcomes compared to non-molecularly oriented therapies. Accelerating the use of genomics could yield a better understanding of the mechanisms of circumvention and resistance to targeted therapies and could increase the chances for obtaining more effective molecular precision medicine for patients with multiple myeloma.

**Abstract:**

Background: Despite that cytogenetic and molecular analysis of tumor cells can rapidly identify recurring molecular abnormalities, no personalized therapy is currently available in the setting of relapsed/refractory multiple myeloma (r/r MM). Methods: MM-EP1 is a retrospective study aimed at comparing a personalized molecular-oriented (MO) versus a non-molecular-oriented (no-MO) approach in r/r MM. Actionable molecular targets and their associated therapies were the BRAF V600E mutation and BRAF inhibitors; t(11;14)(q13;q32) and BCL2 inhibitors; and t(4;14)(p16;q32) with FGFR3 fusion/rearrangements and FGFR3 inhibitors. Results: One hundred three highly pretreated r/r MM patients with a median age of 67 years (range 44–85) were included. Seventeen (17%) patients were treated using an MO approach with BRAF inhibitors (vemurafenib or dabrafenib, *n* = 6), BCL2 inhibitor (venetoclax, *n* = 9), or FGFR3 inhibitor (erdafitinib, *n* = 2). Eighty-six (86%) patients received non-MO therapies. Overall response rate was 65% in MO patients versus 58% in the non-MO group (*p* = 0.053). Median PFS and OS were 9 and 6 months (HR = 0.96; CI95 = 0.51–1.78; *p* = 0.88) and 26 and 28 months (HR = 0.98; CI95 = 0.46–2.12; *p* = 0.98), respectively, in MO and no-MO patients. Conclusion: Despite the low number of patients treated with an MO approach, this study highlights the strengths and weakness of a molecular-targeted approach for the treatment of multiple myeloma. Widespread biomolecular techniques and improvement of precision medicine treatment algorithms could improve selection for precision medicine in myeloma.

## 1. Introduction

Our increased understanding of cancer biology has enabled the identification of molecular alterations that drive cancer progression that has translated into major therapeutic improvements for patients whose cancer harbors targetable molecular alterations. Striking examples include HER2 (human epidermal growth factor receptor−2) amplifications in approximately 15% of breast cancer patients [1] and EGFR (epidermal growth factor receptor) activating mutations in lung cancer, in approximately 15% of western European patients [2]—molecularly oriented treatments for patients with breast cancer and HER2 inhibitors [3] or lung cancer with EGFR inhibitors [4] have considerably improved patient survival. Targeting of other altered oncogenes (e.g., B-Raf proto-oncogene—BRAF, ROS proto-oncogene 1—ROS1, anaplastic lymphoma kinase—ALK, fibroblast growth factor receptors—FGFR, neurotrophic tyrosine receptor kinase—NTRK, Kirsten rat sarcoma virus–KRAS) showed efficiency [5], giving rise to the concept of precision medicine, which aims at treating patients according to the molecular portrait of their tumor.

Most precision medicine trials are designed for solid tumor treatments. Dedicated precision medicine studies, with oriented molecular therapies based on a specific molecular abnormality for multiple myeloma (MM), have not yet been reported.

MM is a neoplastic disorder characterized by clonal proliferation of malignant plasma cells in bone marrow, with a monoclonal immunoglobuline accumulation that can lead to organ dysfunction. Since the mid-1990s, the treatment landscape for MM has dramatically evolved, resulting in improved outcomes and longer survival for patients [6]. The identification of actionable molecular abnormalities in MM such as BRAF mutations, along with the development of BRAF inhibitors, suggests the possibility of patient-tailored treatments targeting clonal plasma cells with a specific oncogenomic profile [7,8,9,10,11]. Venetoclax, a BCL2-inhibitor, has showed meaningful therapeutic responses in relapsed multiple myeloma in patients carrying the t(11;14) (q13;q32) cytogenetic abnormality, reviving the interest in a treatment specifically for this subgroup of patients [12]. Indeed, venetoclax treatment showed 86% of responses among t(11;14) relapsed/refractory (r/r) MM patients, and its benefit has been shown to be restricted to patients carrying this translocation [13,14].

Another potential therapeutic molecular target in MM is fibroblast growth factor receptor−3 (FGFR3), which overexpression correlates with the translocation t(4;14) (p16;q32), present in ~20% of MM patients. FGFR-inhibitors (FGFR3-i), already available in solid malignancies, represent an attractive therapeutic option for patients with t(4;14)-related FGFR3 fusion/rearrangements [15,16].

The development of next-generation sequencing techniques identifying MM patient-specific molecular profile, helps orient patients toward appropriate targeted therapies and accelerates progress towards personalized precision medicine [6,17,18]. Potential other molecular targets in multiple myeloma deserve to be further investigated, such as loss of 1p31 region involving CDKN2C (cyclin dependent kinase inhibitor 2C) gene, potentially targeted by CDK4/6 inhibitors such as asabemaciclib; NRAS and KRAS mutations potentially targeted by MEK inhibitors such as ascobimetinib or other NRAS / KRAS further selective inhibitors; MDM2 (murine double minute 2), potentially targeted by p53-MDM2 inhibitors such as idasanutlin; TP53 (tumor protein 53) abnormalities, potentially targeted by TP53 activators such as APR-246; and IDH2 (isocitrate dehydrogenase-2) mutations potentially targeted by IDH2 inhibitors such as enasidenib [19,20,21].

MM-EP1 is a retrospective study aiming to assess the clinical benefit of a personalized molecular-oriented (MO) approach versus a non-molecular-oriented (no-MO) for patients with r/r MM.

## 2. Materials and Methods

The MM-EP1 study is a retrospective study carried out by analyzing data collected at Gustave Roussy Hospital, Villejuif, France, from 2013 to 2022, about 103 patients with r/r MM after at least one line of treatment. The MM-EP1 study aimed to retrospectively assess a personalized MO approach versus a no-MO approach for the treatment of patients with r/r MM. The working hypothesis was that patients that had been treated in a molecularly oriented way would have a higher overall anti-tumor response rate or a longer progression-free survival.

Molecular screening methods including cytogenetics of tumor plasma cells, Sanger PCR analysis and targeted NGS analysis (29 disease-related genes, Appendix A) on magnetic-sorted CD138-positive bone marrow cells. Cytogenetic results were obtained by performing conventional chromosome banding accompanied by fluorescence in situ hybridization (FISH) realized on CD138-sorted plasma cells. The NGS analysis relied on targeted sequencing using a customized AmpliSeq panel covering exonic coding regions of 29 myeloma-related genes (IAD191489; Ion Torrent). All patients included in the MM-EP1 study with molecular material available for analysis consented to molecular screening on bone marrow aspiration.

From 2013 to 2018, 57 patients underwent molecular screening for BRAF, KRAS, NRAS and TP53 mutations using the Sanger technique. From 2019 to 2022, targeted NGS was performed on a sorted tumor population of 45 patients. In one patient, the molecular analysis technique was not performed due to a lack of tumor DNA.

For analysis, we considered as treated with an MO approach (MO-arm) those patients presenting t(11;14) who had been treated with BCL2 inhibitors such as venetoclax; patients with t(4;14) and with FGFR3 rearrangement treated with FGFR3 inhibitors such as erdafitinib; and those with BRAF mutation, treated with BRAF inhibitors such as vemurafenib or dabrafenib. Patients treated with other anti-myeloma therapies, without a specific molecular orientation, were included in no-MO arm (Figure 1).

Anti-myeloma treatment in both groups (MO and no-MO) could have been administered either in the context of a clinical trial (conditional on slot availability), or according to available reimbursed molecule at the time of treatment, and the choice of treatment for each patient was left to the discretion of the physician treating the patient.

The primary objective of the MM-EP1 study was to evaluate the feasibility and potential benefit of a personalized therapy oriented by the abovementioned target detection (MO-arm) over standard anti-myeloma therapy (no-MO-arm). The tumor response rate, overall survival (OS, defined as the time of myeloma diagnosis to last follow up or death), progression-free survival (PFS) and percentage of reduction in monoclonal component were assessed in both groups. Response to treatment was evaluated using the International Myeloma Working Group Uniform Response Criteria [22]. OS and PFS were calculated using the Kaplan–Maier estimator. Statistical analysis was performed with IBM SPSS statistics v28 and R. All patients gave their written consent for molecular analysis and to be included in the MM-EP1 study. This work was conducted in agreement with the principles of the Helsinki declaration.

## 3. Results

### 3.1. Patients’ Characteristics

One hundred three patients with r/r MM were included in the study. The mean age was 67 years (range 44–85); at inclusion, the median of previous lines of therapies was 4 (range 1–8) (Table 1). Prior systemic therapies included immunomodulatory agents (*n* = 103; 100%), alkylating agents (*n* = 99; 96%) or proteasome inhibitors (*n* = 99; 96%), and 73 (73%) patients had previously received autologous stem-cell transplantation.

### 3.2. Molecular Characteristics of r/r MM

Cytogenetic analysis was successfully obtained for 93 patients (90%); the remaining 10 (10%) patients had too few tumor cells for an accurate cytogenetic analysis. Most recurrent cytogenetic alterations were chromosome 1 abnormalities (*n* = 20; 20%), followed by t(4;14) (*n* = 21; 20%), t(11;14) (*n* = 19; 18%) and del17p (*n* = 12; 12%).

Upon NGS analyses, the median number of mutant genes per patient was 2 (range 1–7). The most frequently mutated genes were KRAS (53%), NRAS (27%), family with sequence similarity 46, member C- FAM46C (24%), DIS3 (16%), TP53 (13%), MYC associated factor X–MAX (9%), BRAF (9%), Splicing Factor 3b Subunit 1–SF3B1 (7%) and ten-eleven-translocation 2 -TET2 (7%) (Figure 2); variant allele frequency (VAF) for each detection is shown in Appendix A. The majority of NRAS and KRAS mutations were substitution at codon 61 (92% and 29%, respectively, for NRAS and KRAS), followed by codons 12 and 13 (8% and 57%, respectively, for NRAS and KRAS). The most common mutation in BRAF was the V600E substitution (2 out of 4 patients) (Appendix A). The mutational interaction matrix shows a significantly co-occurrence of NRAS and FAM46C mutations (*p* < 0.01) and the significant mutual exclusion of KRAS and NRAS mutations (*p* < 0.05) (Appendix A).

Correlations between cytogenetic abnormalities and mutational profiles are shown in Figure 3. DIS3 and KRAS mutations were mainly associated with a chromosome 1 abnormality. BRAF and NRAS mutations were associated with a normal karyotype. Only a fraction of patients with del17p had a TP53 gene mutation, and conversely only a fraction of patients with the TP53 mutation had del17p on the karyotype (Figure 3).

### 3.3. Potentially Actionable Molecular Target Identification

A potentially actionable target as defined by the MM-EP1 study criteria (Figure 1) was identified in 47 of the 103 (48%) patients. These molecular abnormalities were t(4;14) with FGFR3 fusion/rearrangements (21 patients, 20%), t(11;14) (19 patients, 18%) and BRAF V600E mutations (7 patients, 7%).

### 3.4. Molecularly Oriented and Treated Patients

Although potentially targetable molecular abnormalities were identified in 47 patients, 17/47 patients were finally included in an early phase clinical trial (phase 1–2) or early drug access program for molecularly oriented treatment. Patients not fulfilling the criteria for the protocol or the program, a lack of available treatment slots in the clinical trial, or the decision of the patient or the referring physician were reasons for not conducting molecularly oriented therapy.

Finally, 17 (17%) of the 103 patients were treated with personalized MO therapies and 86 (84%) of the 103 patients were treated with no-MO therapies. Patients treated with personalized MO therapies received a BRAF inhibitor-based therapy (vemurafenib *n* = 3; dabrafenib, *n* = 3), a BCL2 inhibitor-based therapy (venetoclax, *n* = 9), or a FGFR3 inhibitor (erdafitinib, *n* = 2). The details of MO therapies received by patients are summarized in Table 1.

### 3.5. Efficacy Endpoints

All 103 patients included in the MM-EP1 study received treatment for their myeloma following the molecular and cytogenetic screening. Considering all 103 multiple myeloma treated patients, the overall response rate (ORR) was 60%, and 29% of patients achieved at least a very good partial response (VGPR), defined as a > 90% reduction in serum M-protein (Table 2, Figure 4). The median variation in measurable monoclonal component in serum was a decrease of 34%. Details of molecular-oriented patients and correlation with response assessment are reported in Table 3.

In MO-treated patients, ORR was 65% (54% with at least VGPR) and in no-MO patients the ORR was 58% (24% with at least VGPR) (*p* = 0.053) (Figure 4). The median variation in measurable monoclonal component in serum was a decrease of 91% in MO-treated patients and decrease of 30% in no-MO treated patients (*p* = 0.33) (Figure 5).

The median PFS was 9 and 6 months, respectively, in MO and no-MO treated patients (HR = 0.96; CI95 = 0.51–1.78; *p*= 0.88). Median OS was 26 and 28 months, respectively, in MO and no-MO treated patients (HR = 0.98, CI95 = 0.46–2.12; *p* = 0.98) (Table 2, Figure 6). At last follow-up, 54 deaths occurred: 8 in the MO-arm and 46 in the no-MO-arm. Causes of death were MM progression (*n* = 47), secondary malignancies (*n* = 3), COVID-19 (*n* = 2), 1 death due to influenza infection (*n* = 1) and heart coronary ischemic disease (*n* = 1).

## 4. Discussion

To the best of our knowledge, the MM-EP1 study is the first study to compare the outcome of patients with r/r MM treated with molecularly and non-molecularly oriented treatments. The MO treatment approach showed similar overall response rate, PFS and overall survival as compared to non-molecularly oriented patients. The endpoint of the study was not reached, as the benefit of patients receiving molecularly oriented treatment was not improved according to the selected parameters of response rate and PFS.

Consistent with data reported in other studies [23], we found that the mostly recurrent mutations in r/r MM were KRAS, NRAS, FAM46C, DIS3 and BRAF. Our review panel reports co-occurring mutations for NRAS and FAM46C, and other mutually exclusive mutations such as KRAS and NRAS, as previously reported [7,24]. Interestingly, in our study, the coexistence of different mutations of the MAPK pathway was observed in two patients with a NRAS/KRAS/BRAF mutated (VAF: 2%, 10% and 7% and 26%, 39% and 9%), one case NRAS/KRAS mutated (VAF: 39% and 2%) and one case KRAS/BRAF mutated (VAF: 66% and 4%). The coexistence of different mutations probably correlates with an important clonal heterogeneity in heavily pre-treated r/r MM patients (i.e., patients with NRAS/KRAS/BRAF mutations received five and four previous lines of treatment).

The benefit of molecularly oriented cancer treatment and the magnitude of this benefit remains a matter of debate. Therapeutic molecular precision medicine programs have so far been developed primarily in oncology for the treatment of solid tumors. The results of the SHIVA01 trial, the first prospective, randomized precision medicine trial comparing targeted therapy based on the molecular profile of the tumor with treatment by physician’s choice in patients with various types of metastatic cancer, was negative for its primary endpoint (i.e., progression-free survival). The results of SHIVA01 did not confirm the hypotheses generated by non-randomized clinical studies, including the pilot study by von Hoff et al. [13], the MD Anderson Cancer Center experiment [14,15] and the MOSCATO trial [16]. The potential benefit of receiving a molecularly oriented treatment was then generally limited to a small proportion of patients (7% of patients included in the MOSCATO study, for example).

The difficulty in generating benefits with precision medicine clinical trials can be explained by several reasons. First, there is a limited number of targeting agents available [25]. Second, most of the treatment algorithms for the precision medicine molecular clinical trials used are one-dimensional (i.e., one drug for one molecular target) and do not consider resistance mechanisms (e.g., BRAF or KRAS mutations) [13,26,27]. Third, the genetic characteristics and heterogeneity of tumor cells may be responsible for treatment failure [23,28]. The absence of benefit in terms of duration of response in our MM-EP1 study may be related to clonal heterogeneity, present in a majority of studied samples and known to be an important hallmark of advanced and multi-treated multiple myeloma [7,23]. In the same way, drug efficacy may depend on coexisting molecular alterations. For example, patients whose solid tumors harbor a phosphatidylinositol 3-kinase (PI3KCA) mutation respond primarily to PI3K inhibitors if they are devoid of a concomitant KRAS mutation [29]. Incorporating information about two molecular alterations to guide toward a specific drug is a two-dimensional processing algorithm and could improve the results of molecular precision medicine trials. Considering the increasing number of oncogenic molecular anomalies discovered in recent years, the complexity of algorithms is likely to increase and artificial intelligence could be of precise assistance in helping to choose the most appropriate treatment for a given patient [30].

The molecular and cytogenetic data provided in our study increase the knowledge of the potential relationships between genetic mutations and karyotype abnormalities in r/r MM patients. For example, only a fraction of patients with del17p had a concomitant TP53 gene mutation based on our NGS panel, and conversely only a fraction of patients with the TP53 mutation had del17p on the karyotype, suggesting that neither technique, taken singularly, can provide complete and affordable information on TP53 integrity. Our study suggests that molecular biology and karyotype provide potentially complementary information for the evaluation of r/r MM.

Phase II studies have shown the efficacy of BRAF inhibition in multiple myeloma with BRAF V600E mutation [26]. However, the response durations were short and almost all patients lost their therapeutic response within the 3 months of treatment, suggesting the interest in combining targeted therapies involved in MAP (mitogen-activated protein) kinase pathways [26]. The combination of BRAF inhibitor with MEK inhibitor [11,31,32,33] is currently being studied in patients with r/r MM (CAPTUR trial, NCT03297606; ROAR trial NCT02034110).

Furthermore, the MAPK pathway could be targeted with new therapeutics, such as specific inhibitors of KRAS G12C mutations that have recently demonstrated meaningful anti-cancer activity in lung cancer with KRAS G12C mutations. In multiple myeloma, KRAS mutations are substitutions at codon 61 for which, to our knowledge, there is no clinically available specific inhibitor. Pre-clinical results for AZD4785, a new drug that functions as an antisense oligonucleotide that downregulates all KRAS isoforms, has shown inhibition of myeloma cell growth with KRAS mutations [15].

Translocation t(4;14) may activate the FGFR3 oncogene by gene fusion or rearrangement, and some preclinical data show that cell growth arrest and apoptosis in myeloma cell lines bearing this fusion/rearrangement can be achieved by FGFR3 inhibition [24]. In our study, we treated two patients with a FGFR3 inhibitor (erdafitinib; early access program), and these two patients did not achieve an objective therapeutic response. Future studies are warranted to further explore the effect of FGFR3 inhibition in multiple myeloma with t(4;14) and a better understanding of the implications of FGFR3 oncogenic partners and other partners like MMSET that play a preeminent oncogenic role in patients with t(4;14) [34,35].

Other targeted- or tumor biology-oriented therapies could be worth investigating in future studies. BCMA (B-cell maturation antigen)-targeted therapies, with chimeric antigen receptor (CAR) T-cells and antibody–drug conjugates (belantamab–mafadotin), and CD3 x anti-BCMA bi-specific antibodies now raise the question of BCMA screening to better identify candidate and selected patients in such appropriate therapies [36]. Other actionable molecular targets have been suggested to treat MM, such as isocitrate dehydrogenase (IDH)1-IDH2 mutations with IDH specific inhibitor [37]. MCL-1(myeloid cell leukemia sequence 1) overexpression on tumor plasma cells, associated with amplifications of chromosome 1q21, could be a possible therapeutic target with MCL-1 inhibitors [38]. Finally, targeting and restoring TP53 functions or targeting MDM2 with specific inhibitors in cases of myeloma with 17p deletions could also be interesting treatment avenues to investigate [11,12].

Moreover, from a theoretical point of view, because MM is a genetically heterogenic disease, an approach combining MO and no-MO drugs could also provide interesting results. In fact, these newly created triplet-quadruplets may allow synergic and diversified activity on myeloma cells, producing deeper and more prolonged response.

Concerning risk assessment in MM, easier availability of molecular analysis could help in identifying those patients who may take advantage of an MO-approach.

Finally, if the new phase 3 study (such as NCT03539744 with venetoclax) meets its objectives and confirms the results, BCL2 inhibitors should soon be able to be used in patients with relapsed multiple myeloma with t(11;14), and would then be the first targeted therapy to be approved on a biomarker in the myeloma field. In our study, venetoclax showed to be highly active in t(11;14) carrier patients, and only two patients, both carrying chromosome 1 abnormalities too, progressed under this treatment. Additionally, patients should be considered for molecularly stratified treatment options for some very specific molecular abnormalities, such as those of the BRAF and RAS/RAF pathway, as specific inhibitors are already available, and others are in development in oncology for other indications. Finally, we must recognize that, considering the promising new treatment options targeting CD38, SLAMF7 (SLAM family member 7), BCMA, most patients will be able to receive therapies that are not specifically molecularly oriented. Patients would then potentially be referred to treatment based on non-molecular biomarkers, such as immunological or cell surface biomarkers.

We acknowledge several limitations in our work, including non-randomized identification and inclusion of patients, and the retrospective nature of the study. Selection biases of patients may have occurred. The orientation decision, which is obviously limited to the availability of other therapeutic opportunities, is another potential bias. Finally, the sequencing panel includes only 29 genes and does not allow a complete characterization of the r/r MM samples. Owing to these different limitations and the limited number of patients included, we cannot draw definitive conclusions about the potential benefit of precision medicine in multiple myeloma treatments. However, our study demonstrates the feasibility and underscores strengths and weaknesses of routine genomic characterization in patients with r/r multiple myeloma.

## 5. Conclusions

As compared to treatments without molecular guidance, molecularly oriented therapies for relapsed or refractory multiple myeloma patients were not translated into improved outcomes. Accelerating the use of genomics and improving precision medicine treatment algorithms could increase the chances for achieving effective molecular precision medicine for patients with multiple myeloma and could open the way to new interesting drug combinations.

## Figures and Tables

**Figure 1 cancers-15-01508-f001:**
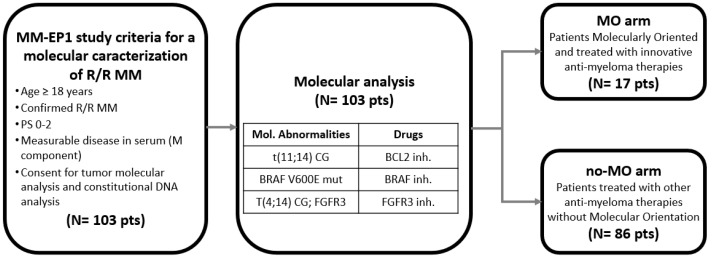
Flow chart of the MM-EP1 study. The treatment algorithm for precision medicine was based on molecular abnormalities: t(11;14) chromosomal translocation therapeutically targeted by a BCL2 inhibitor; t(4;14) chromosomal translocation with fusion/FGFR3 rearrangement targeted by a FGFR3 inhibitor; and the *BRAF* V600E mutation targeted by a BRAF inhibitor. r/r MM: relapsed/refractory multiple myeloma, PS: performance status, MO: molecularly oriented, no-MO: no molecularly oriented.

**Figure 2 cancers-15-01508-f002:**
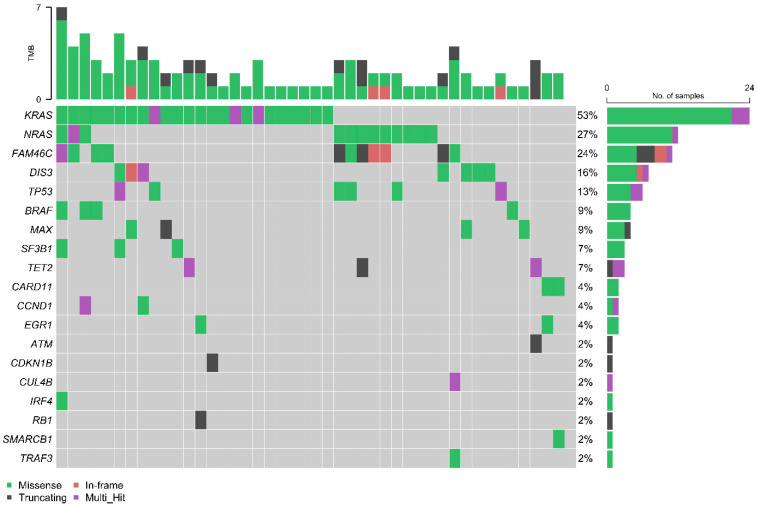
Oncoplot of altered genes in sorted CD138^+^ myeloma cells in patients with relapsed or refractory multiple myeloma. The target NGS analysis was performed on 45 patients.

**Figure 3 cancers-15-01508-f003:**
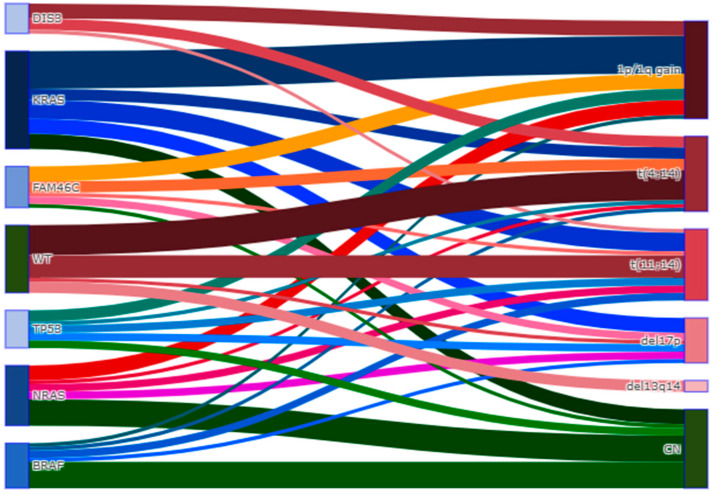
Sankey diagram showing the relationships between the six most frequently mutant genes *DIS3*, *KRAS*, *FAM46C*, *TP53*, *NRAS* and *BRAF* (**left**) and the cytogenetic characteristics (**right**), in patients with relapsed or refractory multiple myeloma.

**Figure 4 cancers-15-01508-f004:**
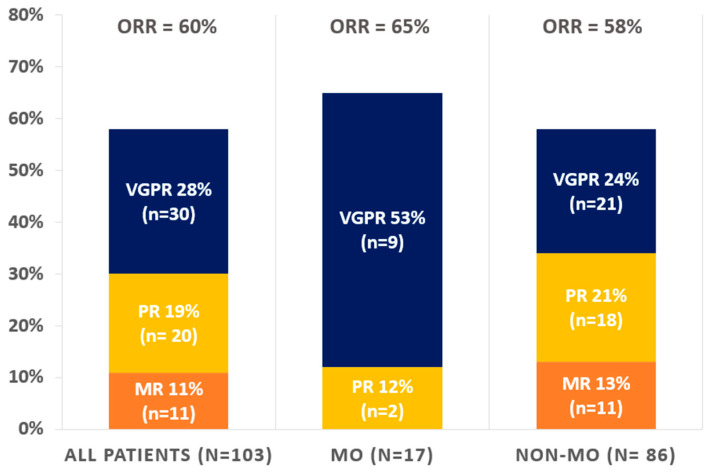
Overall response rate obtained in patients treated for relapsed or refractory multiple myeloma, according to a molecularly oriented (MO) or non-molecularly oriented (no-MO) treatment. MO: molecularly oriented, NON-MO: not molecularly oriented, ORR: overall response rate, VGPR: very good partial response, PR: partial response, MR: minimal response.

**Figure 5 cancers-15-01508-f005:**
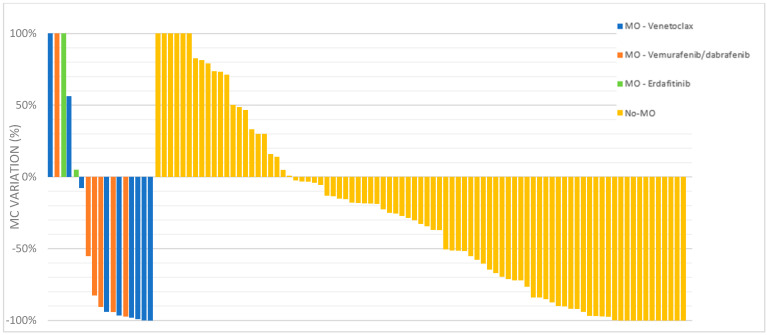
Change in monoclonal component serum measurement according to MO or no MO treatment in patients with R/R MM included in the MM-EP1 study. A cascade plot of the percent change in serum monoclonal component was made as a function of the best treatment response (nadir) achieved during treatment. MC: monoclonal composant, MO: molecularly oriented, VEN: venetoclax, BRAFi: BRAF inhibitor, FGFR3i: FGFR3 inhibitor.

**Figure 6 cancers-15-01508-f006:**
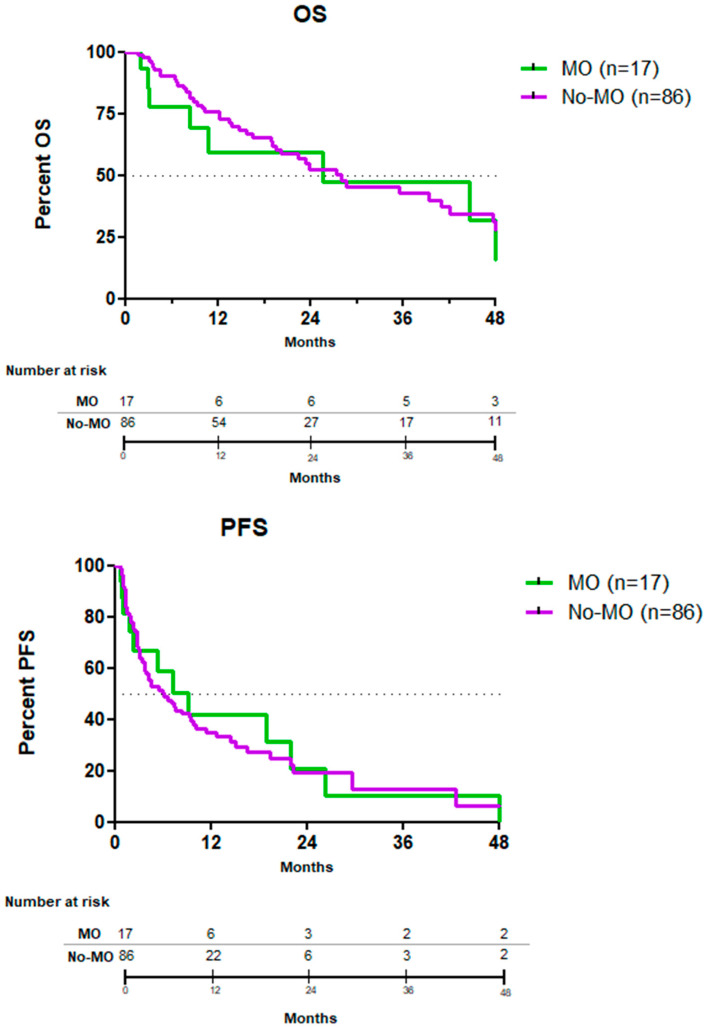
Kaplan–Meier plots of overall survival and progression-free survival by MO and no-MO treatments in patients with relapsed or refractory MM included in the MM-EP1 study. MO: molecularly-oriented, No-MO: not molecularly oriented, OS: overall survival, PFS: progression-free survival, MM: multiple myeloma.

**Table 1 cancers-15-01508-t001:** Descriptive characteristics of the cohort and two treatment subgroups.

	All Patients (*n* = 103)	MO (*n* = 17)	No-MO (*n* = 86)
Number, M/F	102, 56/47	17, 7/10	86, 49/37
Median age [range]	67 yo [44; 85]	59 yo [48; 81]	68 yo [44; 85]
Involved chain			
IgG	60% (*n* = 62)	41% (*n* = 7)	63% (*n* = 54)
IgA	17% (*n* = 18)	24% (*n* = 4)	16% (*n* = 14)
FLC	23% (*n* = 24)	35% (*n* = 6)	21% (*n* = 18)
Kappa	17% (*n* = 18)	29% (*n* = 5)	15% (*n* = 13)
Lambda	6% (*n* = 6)	6% (*n* = 1)	6% (*n* = 5)
ISS			
I	28% (*n* = 29)	29% (*n* = 5)	28% (*n* = 24)
II	22% (*n* = 23)	41% (*n* = 7)	17% (*n* = 15)
III	23% (*n* = 24)	12% (*n* = 2)	26% (*n* = 22)
NA	27% (*n* = 28)	18% (*n* = 3)	29% (*n* = 25)
Cytogenetic risk			
t(4;14)	20% (*n* = 21)	24% (*n* = 4)	20% (*n* = 17)
t(11;14)	18% (*n* = 19)	53% (*n* = 9)	12% (*n* = 10)
chr1 abnorm	19% (*n* = 20)	24% (*n* = 4)	20% (*n* = 17)
del13q14	3% (*n* = 3)	6% (*n* = 1)	2% (*n* = 2)
del17p	12% (*n* = 12)	18% (*n* = 3)	11% (*n* = 9)
NA	11% (*n* = 11)	12% (*n* = 2)	11% (*n* = 9)
Number prior lines median [range]	4 [1–8]	5 [1–11]	3 [1–8]
Prior IMID	100% (*n* = 103)	100% (*n* = 17)	100% (*n* = 86)
Prior alkylating agent	96% (*n* = 99)	100% (*n* = 17)	95% (*n* = 82)
Prior PI	96% (*n* = 99)	100% (*n* = 17)	95% (*n* = 82)
Prior dara (anti-CD38)	23% (*n* = 24)	58% (*n* = 6)	21% (*n* = 18)
HSCT auto/allo	72%/6% (*n* = 73/6)	69%/19% (*n* = 11/3)	72%/4% (*n* = 62/3)
Treatment considered (no. of patients)		BRAFi-based therapy: vemurafenib (3) dabrafenib (3) BCL2i-based therapy: venetoclax (9) FGFR3i-based therapy: erdafitinib (2)	Anti-CD38 (27) IMID-based (34) CPI (13) Belantamab (8) PI (20) MDM2 inhib (2) Cobimetinib (3) mTOR (1) Isatuximab (3) Other (1)

MO: molecularly oriented, No-MO: not molecularly oriented, M: males, F: females, yo: years old, Ig: immunoglobulin, FLC: free light chain, ISS: International Staging System, NA: not available, t: translocation, chr: chromosome, del: deletion, IMID: immunomodulatory drugs, PI: proteasome inhibitors, Dara: daratumumab, HSCT: hematopoietic stem cell transplantation. CPI, check-point inhibitor.

**Table 2 cancers-15-01508-t002:** Outcome data in the whole population and subgroups. MO: molecularly oriented.

	All Patients (*n* = 103)	MO (*n* = 17)	No-MO (*n* = 86)	*p*-Value
Time to best response	7 ± 1 months	9 ± 4 months	6 ± 1 months	0.47
Response				
VGPR	28% (*n* = 30)	53% (*n* = 9)	24% (*n*= 21)	0.053
PR	19% (*n* = 20)	12% (*n* = 2)	21% (*n* = 18)	
MR	11% (*n* = 11)	0% (*n* = 0)	13% (*n*= 11)	
SD	21% (*n* = 22)	12% (*n* = 2)	23% (*n* = 20)	
PD	20% (*n* = 21)	24% (*n* = 4)	20% (*n*= 17)	
Δ MC (median)	−34%	−91%	−30%	0.33
PFS (median)	7 months	9 months	6 months	0.88
OS (median)	28 months	26 months	28 months	0.98
Deaths	52% (*n* = 54)	47% (*n* = 8)	53% (*n* = 46)	0.23
MM-related	85% (prop: 45/54)	88% (prop: 7/8)	85% (prop: 39/46)	

No-MO: not molecularly oriented, NS: no significant difference, VGPR: very good partial response—>90% reduction in serum M-protein, PR: partial response—>50% reduction in serum M-protein, MR: minimal response—M protein reduction between 25% and 50%, SD: stable disease, PD: progressive disease—M protein augmentation >25%, Δ MC: variation in monoclonal component, PFS: progression-free survival, OS overall survival, MM: multiple myeloma.

**Table 3 cancers-15-01508-t003:** Details of molecular-oriented patients correlating with response assessment.

Pt No°	Involved Chain	Cytogenetic/Molecular Profile	Therapy	M-Composant Variation
1	κ	t (4;14); t (11;14); chr. 1 abn; Δ17p; TP53 mut	VEN-based	+882
2	IgG, κ	BRAF mut V600E	BRAFi based (vemurafenib)	+296
3	IgG, λ	t (4;14)	FGFRi based (erdafitinib)	+181
4	λ	t (11;14); chr. 1 abn	VEN-based	+56
5	IgG, κ	t (4;14)	FGFRi based (erdafitinib)	+2.3
6	IgG, κ	t (11; 14); BRAF mut G464E; NRAS mut Q61R	Ven-based	−8
7	IgA, κ	BRAF mut V600E	BRAFi based (dabrafenib)	−55
8	IgA, κ	BRAF mut V600E	BRAFi based (vemurafenib)	−83
9	IgG, λ	t (11;14); BRAF mut V600E	BRAFi based (dabrafenib)	−91
10	IgA, κ	t (4;14); t (11;14)	VEN-based	−94
11	IgG, λ	BRAF mut V600E	BRAFi based (vemurafenib)	−94
12	κ	t (11;14)	VEN-based	−97
13	IgA, λ	BRAF mut V600E	BRAFi based (Dabrafenib)	−97
14	κ	t (11;14); chr. 1 abn; Δ17p	VEN-based	−98
15	κ	t (11;14)	VEN-based	−99
16	κ	t (11;14)	VEN-based	−100
17	IgG, λ	t (11;14)	VEN-based	−100

Pt: patient, M: monocolonal, t: translocation, κ: kappa chain, λ: lambda chain, VEN: venetoclax, BRAFi: BRAF inhibitor, FGFR3i: FGFR3 inhibitor.

## Data Availability

The data that support the findings of this study are available from the corresponding author upon reasonable request.

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
