# Peer review of "Precision Medicine Approach Based on Molecular Alterations for Patients with Relapsed or Refractory Multiple Myeloma: Results from the MM-EP1 Study"

_cancers, 2023, doi:10.3390/cancers15051508_

Round 1

Reviewer 1 Report

1.      The study design of this study is somewhat confused. In the Materials and Methods, the authors describe that the MM-EP1 was phase 1 to 2 clinical trial and aimed to assess a personalized molecular oriented (MO) approach and a no-MO approach. But in line 99, they analyzed the data retrospectively. They performed the mutational analysis before inclusion to the MM-EP1 study to suggest molecular targeted therapy. In addition, in line 178, they say that some patients with early drug access programs were included in the study. Therfore, I couldn’t understand clearly the design of this study.

2.      In the MO group, most frequent mutations or cytogenetic alterations were t(11;14), which is known to be relatively favorable alterations and good response to BCL2 inhibitor. I think that the higher response rates in the MO group in the figure 4 might be the relatively favorable cytogentic alterations. But how do you explain no difference between two groups with regards to PFS?  

3.      Line 107: the total number of patients included in the study differs from the figure 1. 56 patients with Sanger sequencing and 45 with targeted NGS totals 101 patients.

4.      Line 113: Is there any patient with multiple actionable targets? Especially, the combination of mutations by NGS and cytogenetics such t(11;14) and t(4;14). The cytogenetic changes is relatively early events in the development or progression MM, while additional mutations acts as secondary events. If so, what is the selection guideline for the targeted therapy?

5.      It seems that Venetoclax and BRAF inhibitor showed good response but FGFR inhibitor didn’t have any effecacy in figure 5. It helps understant the response to each targeted therapy if all the mutational profiles and cytogenetic fusions of the patients treated with molecular-targeted therapy are described or showed in the figure.

Reviewer 2 Report

Title: Precision medicine approach based on molecular alteration for patients with relapsed or refractory multiple myeloma: results from MM-EP1 study 

In their manuscript ‘Precision medicine approach based on molecular alteration for patients with relapsed or refractory multiple myeloma: results from MM-EP1 study’ Andreozzi et al. report on molecularly stratified treatment strategies a subgroup of 17 patients with relapsed/refractory multiple myeloma. I enjoyed reviewing this manuscript which is well written and of potential clinical importance. There are some minor points that need to be addressed prior to publication. 

Major:

Abstract:

-       The authors concluded that precision medicine can improve r/r MM-patients outcome. This seems to be misleading as they presented similar effectiveness of molecularly stratified treatment compared to non-molecularly stratified treatment options. I believe the authors wanted to point out that further improvements and a widespread availability of molecular diagnostics in daily routine are warranted to increase the efficacy of precision medicine approaches in this setting. 

Minor:

Methods:

-       Lines 112 – 113: Hopefully, the molecular tumor board comprised more specialties than a hematologist, a biologist and clinical trial investigators. Please comment on this sentence. 

-       As both subgroups (MO vs. No-MO) differ regarding the number of patients, it would be interesting to perform a propensity score matching analysis. 

Results:

-       Figure 2: The oncoplot should be renewed including a clear delineation of each case. To me the summary of some cases does not make any sense here.

-       Figure 3: This Figure is very useful. Perhaps the authors are able to color the flows allowing the reader to follow each flow to the cytogenetic alteration (https://r-charts.com/flow/sankey-diagram-ggplot2/). 

-       Table 1: The authors provide interesting and heterogeneous treatment characteristics for the no-MO subgroup. Did the authors include patients from others studies within the no-MO subgroup? If yes, the authors have to point this out in the methods section. Additionally, was cobimetinib given in r/r MM patients harboring BRAF V600E mutations or irrespective of BRAF mutational status? 

Discussion:

-       The discussion is well written and clearly structured. However, the authors should discuss the significance of their results in more detail and give information on the prioritization between no-MO targeted treatment (anti CD38, antiBCMA, SLAMF7) and MO targeted treatment. Which clientele of patients should be considered for molecularly stratified treatment options in the era of several promising novel treatment options in multiple myeloma?

Supplement: 

-       The authors should provide a summary of genes included in the panel for targeted amplicon-based sequencing. 
